coastal hazards; risk reduction; storm impacts; warning levels

**Corresponding author:**
Oscar Ferreira;
Email: oferreir@ualg.pt

# A review of early warning systems for storm-induced coastal flooding and erosion on wave-dominated open coasts

Oscar Ferreira 

FCT, CIMA, Universidade do Algarve, Portugal

## Abstract

Early warning systems for coastal erosion and flooding are currently primarily designed for local applications, offering high-resolution, site-specific predictions. Only a few early warning systems (EWS) are used at large regional or national scales. There is also a lack of standardised indicators and thresholds, which vary widely across systems and hinder cross-regional applicability. While current EWS perform well in binary hazard detection (Yes/No hazard; 80–95% accuracy), they struggle to accurately classify intermediate hazard levels. A lack of comprehensive field datasets has impeded rigorous validation for most systems, with many assessments relying on qualitative observations. Improving the reliability of the EWS requires improving their validation against field data obtained during storms and regular updating of the topobathymetric data to include the actual pre-storm morphology. Currently, most EWS rely on outdated or synthetic morphological inputs, which increases prediction uncertainty. The computational constraints of physics-based models may prevent warnings from being issued in time and have led to the adoption of surrogate approaches that depend on robust training datasets. Furthermore, most systems focus solely on hazard detection, paying limited attention to the risk to assets or populations. Future development must prioritise stakeholder engagement and the co-design of systems that incorporate both hazard and risk assessments, in order to improve their usefulness and facilitate decision-making by end users.

## Impact statement

This review highlights the recent improvements to, and limitations of, early warning systems for storm-induced coastal erosion and flooding on wave-dominated coastlines. While current systems offer accurate, site-specific hazard alerts, they often lack broader regional coverage and struggle to define intermediate hazard levels. Expanding EWS to operate at larger scales would help to identify vulnerable hotspots, especially in data-poor regions, enabling more targeted local responses.

The study also emphasises the importance of standardising hazard indicators to allow consistent comparisons across regions. Integrating real-time data from technologies like drones, beach cameras and satellite imagery could significantly improve prediction accuracy. Moreover, artificial intelligence shows promise in speeding up hazard forecasts, but its success depends on access to robust field data – which remains limited.

Importantly, the research calls for a shift in focus from hazard detection alone to systems that also evaluate risks to people and infrastructure. This requires active collaboration with communities and stakeholders in order to design warnings that are timely, meaningful and actionable. By addressing these gaps, the research paves the way for more inclusive, efficient and scalable coastal hazard management strategies that can better protect lives and livelihoods.

## Introduction

Coastal flooding and erosion induced by storms are notable natural hazards which are projected to intensify due to climate change, primarily through sea level rise and increased storminess (Jongman et al., 2012; van Dongeren et al., 2018; Almar et al., 2021; Jevrejeva et al., 2023; Mentaschi et al., 2023; Souto-Ceccon et al., 2025). In addition to hosting significant populations, coastal zones also contain critical infrastructure and invaluable ecosystems that can be impacted by storm-induced hazards. Thus, the projected increase in coastal erosion and flooding will have substantial socio-economic and ecological consequences, as storms threaten lives, cause damages to developments and infrastructure, and result in loss of ecosystem services. Over 10% of the global population reside in the low-elevation coastal zone (less than 10 m above sea level and hydrologically connected to the coast), with an estimated 128–171 million persons potentially exposed to episodic coastal flooding (Neumann et al., 2015; Kirezci et al., 2020; Athanasiou et al., 2024). Due to climate

change and the growing concentration of people and assets in vulnerable coastal regions, the impact of storms may increase in terms of both number and severity for several coastal areas (Basher, 2006; Jongman et al., 2012; Neumaan et al., 2015; van Dongeren et al., 2018). For example, it is projected that sea level rise will at least double the frequency of coastal flooding in the most exposed locations worldwide during the 21st century (Vitousek et al., 2017; Almar et al., 2021). If no new coastal defences or adaptation measures are implemented, the population potentially exposed to episodic coastal flooding under the RCP8.5 scenario is projected to increase by 52% (Kirezci et al., 2020). Increases in storminess and the intensification of climatic phenomena, such as the El Niño Southern Oscillation, will lead to increased coastal erosion, which could result in irreversible changes to coastal areas (Mentaschi et al., 2018).

Although natural hazard events cannot be prevented, their impact on people and property can be reduced if accurate information is provided to people and stakeholders in a timely manner (Doong et al., 2012; Bogaard et al., 2016). The United Nations (see WMO, 2022a) has defined early warning systems (EWS) as integrated systems of hazard monitoring, forecasting and prediction, disaster risk assessment, communication and preparedness activities and processes that enable individuals, communities, governments, businesses and others to take timely action to reduce disaster risks in advance of hazardous events. According to the Early Warnings for All Initiative (EWAI; WMO, 2022a), they should be based on four fundamental pillars:

Pillar 1: Disaster Risk Knowledge;
Pillar 2: Observations and Forecasting;
Pillar 3: Dissemination and Education;
Pillar 4: Preparedness and Response.

EWS are therefore a vital component of disaster risk reduction strategies, playing a crucial role in managing the increasing risk of storm-induced coastal flooding and erosion. They can forecast coastal hazards and associated risks in advance, providing warnings based on the expected magnitude and geographic extent of the impacts. This enables exposed individuals and authorities to take preventive action and enhances response capabilities in the aftermath of the events (Harley and Ciavola, 2013; Lerma et al., 2018; van Dongeren et al., 2018; Apecechea et al. 2023). This reduces risks, prevents loss of life, and minimises economic impacts in coastal areas. Furthermore, EWS have a high cost-to-benefit ratio, making them highly effective instruments for minimising the impact of extreme events (Ciavola et al., 2015; Apecechea et al., 2023; Garzon et al., 2023b; Papadimitriou et al., 2024).

The scientific understanding of storm processes and associated predictive capabilities advanced significantly in the end of the 20th century, aided by the implementation of systematic monitoring programmes (Morton, 2002) and the development of new predictive models. During the first decade of the 21st century, the need for an EWS that could provide timely warnings to vulnerable coastal populations based on accurate forecasts of storm-related impacts became apparent (e.g. Pietrafesa et al., 2006; Ji et al., 2010). Lane et al. (2008) developed an EWS for coastal flooding based on empirical equations for calculating overtopping discharge. Around this time, Baart et al. (2009) suggested that the enhanced capabilities of morphodynamic models meant that numerical modelling could be employed as a tool for real-time warning systems in coastal areas. Shortly afterwards, Harley et al. (2011) provided the first insights into developing a real-time EWS for storm-related coastal hazards (dune erosion and overwash) by comparing the predictions for two storm events in northern Italy with measured beach and dune responses. Many others have built on these pioneering studies by testing and calibrating numerical models and/or empirical formulations, and validating their potential as an integral component of an EWS.

An EWS based on indicators of storm impacts requires the following modules and information to be available for a given coastal area (adapted from Morton, 2002; Basher, 2006; Ciavola et al., 2015), in alignment with the four pillars of the EWAI (Figure 1):

a) An observation and database storage module including weather, waves, surge and detailed morphological data (Pillars 1 and 2);
b) A forecasting module for storm parameters and morphological changes, including storm impacts (Pillar 2);
c) A decision support module including indicators and hazard/ risk maps (Pillar 2);
d) A warning module with different warning levels (Pillar 3);
e) A visualisation module, which can be made available online or to dedicated end users (Pillar 3);
f) A preparedness and response module providing recommendations or mandatory actions for the population, managers and stakeholders, according to the risk level (Pillar 4).

The main hydrodynamic-based (i.e. related to waves and sea level) coastal hazards caused by storms on sandy coasts (Figure 2) are overwash, inundation and erosion. In extreme cases breaching may also occur (Plomaritis et al., 2018). Despite significant recent advancements, challenges remain in developing and operationalising EWS for these coastal hazards (Ciavola et al., 2015; Harley et al., 2016). Only a small proportion of the EWS referenced in the international literature are operational and incorporated into integrated strategies (decision support systems) for reducing coastal risk (Garzon et al., 2023b; Turner et al., 2024; Chatzistratis et al., 2025). References to EWS for complex coastal areas or near complex coastal structures, such as ports, are even rarer in the published literature. This further emphasises the current need for EWS that can provide more timely and accurate information on the impacts and risks caused by coastal storms. Another common issue in EWS is the difficulty of translating complex model outputs into clear, actionable information for stakeholders (Harley et al., 2016; Idier et al., 2021; Turner et al., 2024).

This paper reviews existing EWS (developed, prototype and operational) for storm-induced coastal flooding and erosion on wave-dominated open coasts (Figure 2). The spatial scales, building methods and technologies used by the different EWS were analysed. The analysis focused on EWS that address coastal flooding (including overwash and overtopping) and/or erosion, considering the impact of wind-generated waves. EWS that solely consider changes in sea level (i.e. tide + storm surge) were excluded from the analysis. EWS that neglect the wave component when assessing coastal hazards were therefore not considered, as they underestimate or exclude coastal erosion and overwash (Stockdon et al., 2023). EWS developed for estuarine flooding and compound flooding were also excluded from the analysis. Although the reviewed EWS could potentially be useful for specific coastal types, such as coral reef-lined coasts, coastal areas with limited fetch or gravel barriers, their applicability may be limited by coastal processes specific to those areas, which must be acknowledged. In such cases, efforts must be made to adapt and improve the existing EWS (see, for example, Winter et al., 2020, for coral reef-lined coasts).

This review will primarily focus on hazard assessment (Pillars 1 and 2 of the EWAI; WMO, 2022a) and will cover warning,

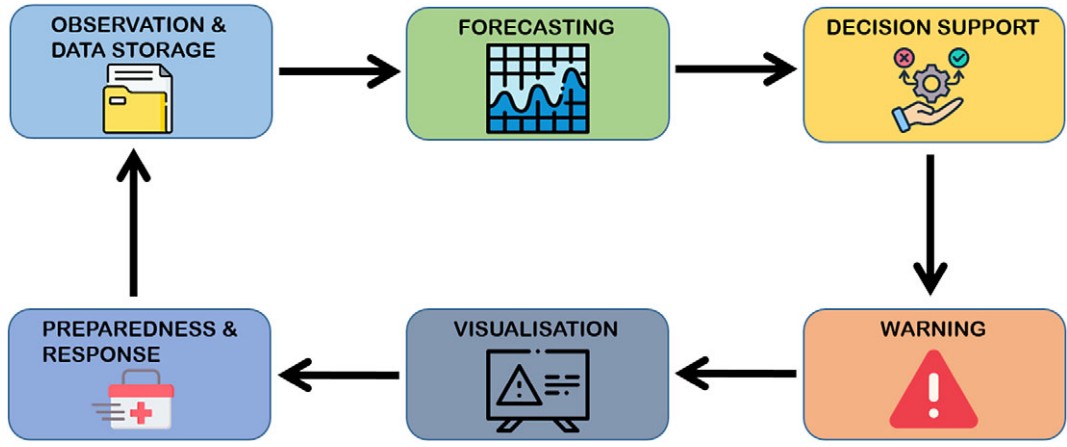

**Figure 1.** The modules required for a fully developed EWS. The small icons in each module were designed by Freepik.

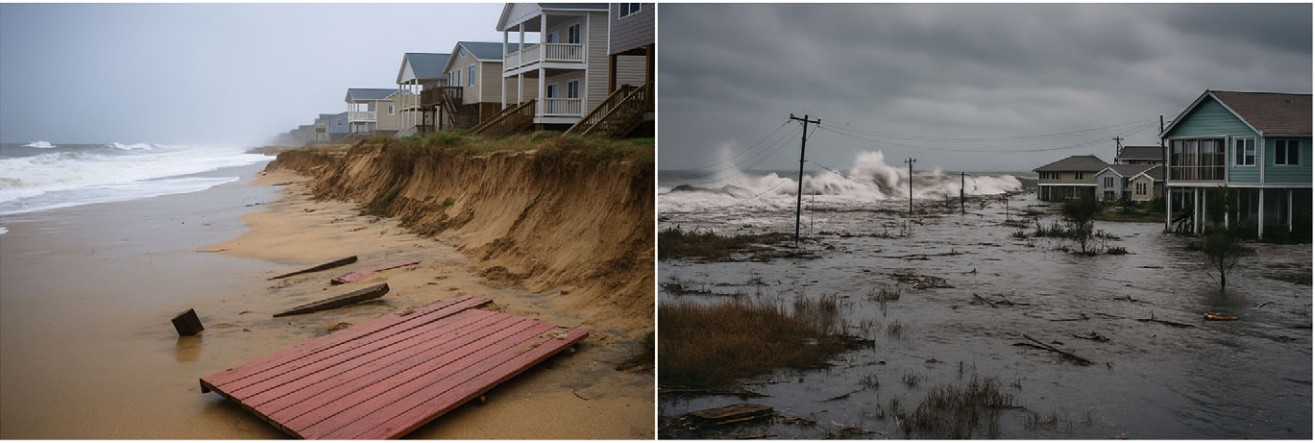

**Figure 2.** Storm-induced coastal erosion (left) and flooding (right), with an illustration of potential associated risks. Images generated using Microsoft Copilot.

communication, preparedness and response actions (Pillars 3 and 4 of the EWAI; WMO, 2022a) to a lesser extent. The analysis covers the considered impact levels and thresholds considered, as well as the estimated accuracy of the developed systems. The EWS review is based almost entirely on published academic literature and does not integrate grey literature or local-national scale reports, which are often written in the official language of each country. Therefore, the analysis may exclude several existing EWS if there is no paper reporting them at an international level. Finally, this paper presents a synthesis of the ongoing challenges and future perspectives on developing EWS for storm-induced coastal hazards.

## Developed early warning systems

EWS can vary greatly in terms of their architecture, models, warning levels, communication type and interaction with stakeholders and end users, resulting in a large number of possible EWS categorisations. One simple way to classify EWS is by their spatial scale of action. It is possible to distinguish two major types of EWS: one aiming for regional/national coverage O(100–1,000 km), typically with lower spatial resolution and relying on simpler predictive tools; and the other for hotspot level O(100–1,000 m), prioritising accuracy and often using more complex physics-based numerical models (Figure 3). In some cases, the same overarching system is designed as a hybrid where both regional/national and local scales coexist (e.g. Turner et al., 2024).

Regional/national-scale EWS (Figure 3a) were developed before local ones (e.g. Lane et al., 2008), but their development remained relatively modest until the early 2020s. Ciavola et al. (2011) identified the difficulties of developing an operational, regional-scale EWS for coastal hazards induced by storms, indicating that this development would take at least 5–10 years from the early 2010s. The authors also indicated that end-users were generally not yet ready to contribute to the development of regional-scale EWS. This was subsequently confirmed by Turner et al. (2024), who stated that, until recently, no national-scale EWS had been operational on sandy coastlines, where storm waves and surges can cause beach erosion and flooding of the beachfront.

The development of EWS prototypes at the hotspot scale (see Figures 3b and 3c) received a significant boost from the European funded MICORE project (Ciavola et al., 2011; Harley et al., 2011; Haerens et al., 2012). This project aimed to set up online EWS using open data feeds in combination with off-the-shelf models to supply predictions up to 3 days before the storm impact (Ciavola et al., 2011). In subsequent years, driven by other seminal projects (e.g. RiscKit; van Dongeren et al., 2018), numerous papers have been published presenting examples of locally developed and tested EWS.

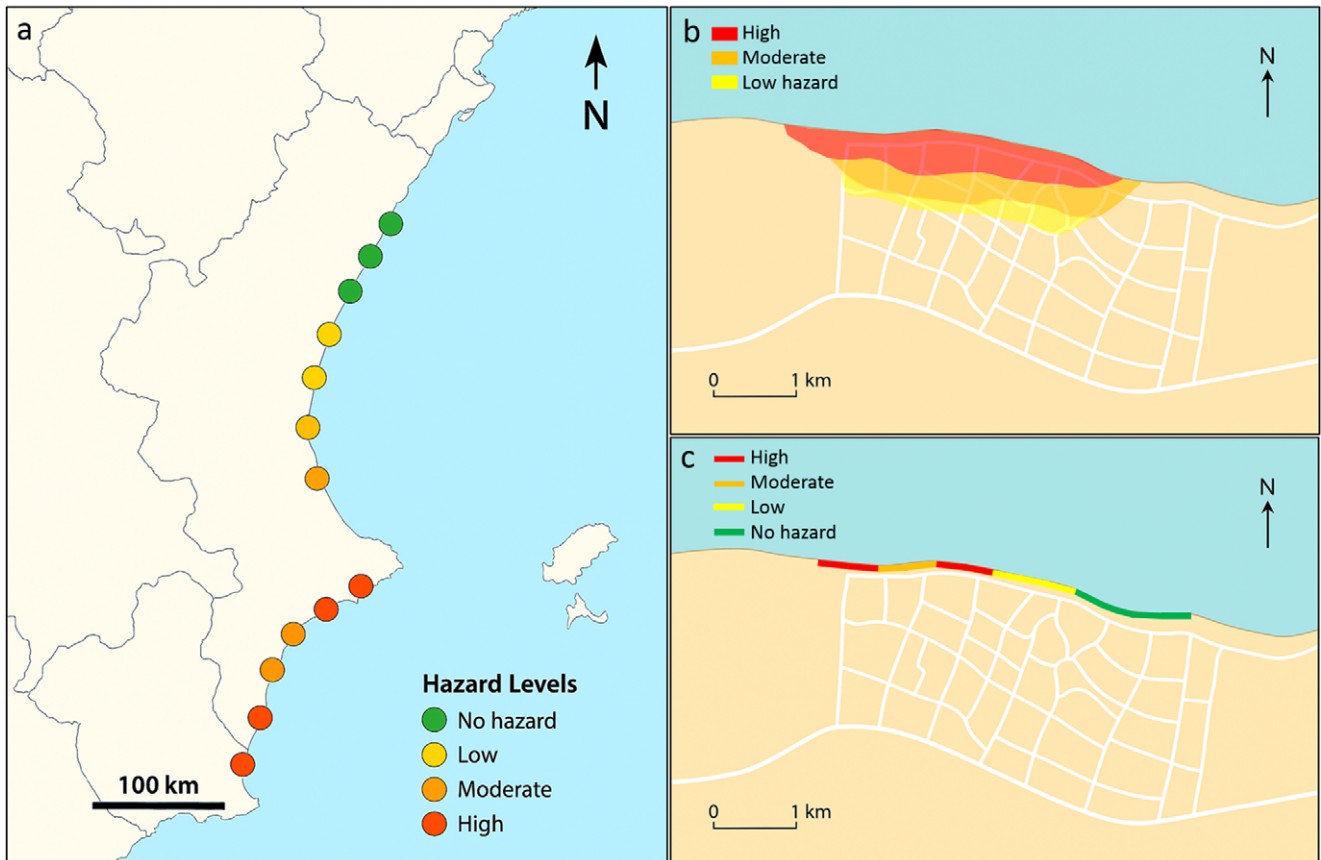

**Figure 3.** Schematic representation of hazard maps: a) Large-scale EWS for storm-induced coastal flooding potential at sandy beaches, b) Local-scale EWS for storm-induced coastal flooding at a coastal village, c) Local-scale EWS for storm-induced coastal erosion at a coastal village. a) Image generated using Microsoft Copilot; b) and c) Base map generated using Microsoft Copilot.

The EWS described here include demonstrators and prototypes (most of which are non-operational), as well as operational systems. It is often difficult to ascertain whether a developed demonstrator or prototype is currently operational and if so, to what extent (e.g. for scientific development, for selected group of end users or fully open). Therefore, that assessment was not performed.

### EWS at the regional/national level

EWS developed at a regional or national scale O(100–1,000 km) mostly rely on the integration of global meteorological models, regional hydrodynamic models (waves, tides and surge) and near-shore hydrodynamic models, as well as parametric formulae, to compute run-up levels (e.g. Stockdon et al., 2006) or overtopping discharge (e.g. Hedges and Reis, 1998; EurOtop, 2018) (Table 1). These large-scale EWS were predominantly developed for flooding caused by overwash/overtopping (Lane et al., 2008; Doong et al., 2012; Zou et al., 2013; Barnard et al., 2014; Gaztelumendi et al., 2016; Lerma et al., 2018; Stokes et al., 2019, 2021; Stockdon et al., 2023; Turner et al., 2024; Quataert et al., 2025), with some also incorporating a storm-induced erosion module (Barnard et al., 2014; Lerma et al., 2018; Turner et al., 2024; Quataert et al., 2025) (Table 1). Only one system (Gorski et al., 2025) was developed solely for erosion (dune impact) (Table 1). Many regional/national scale EWS use the Stockdon et al. (2006) formulation to predict run-up and total water levels, which are then compared with the coastal elevations to define potential impacts. While these levels are most commonly used to determine the likelihood and severity of overwash, they have also been used to identify potential dune impact/erosion when used jointly with the Sallenger (2000) storm impact regimes (e.g. Stockdon et al., 2023; Turner et al., 2024). Empirical parameterisations can rapidly provide available hazard indicators with a high spatial resolution O(100 m) over large areas, since they have been extensively tested and widely adopted (Stockdon et al., 2023), making them a valuable tool for EWS.

The first documented use of physics-based models in the design of a regional-scale EWS to define hazards, replacing parametric formulations, refers to the early development of the CoSMoS system (Barnard et al., 2014). In this study, the authors tested the use of XBeach 1D (Roelvink et al., 2009) to define erosion and flooding at a large number of coastal transects spaced 100–200 m alongshore, extending for hundreds of kilometres. Gorski et al. (2025) followed a similar approach, but only for erosion. XBeach is the most widely used physics-based model for defining erosion and dune impact in large-scale EWS, primarily in 1D mode. In such cases, the model is used repeatedly as if it was applied to a large set of hotspots (hundreds to thousands). In some cases, the modelling is limited to pre-identified critical areas, according to the storm conditions (e.g. Stokes et al., 2019, 2021; Turner et al., 2024). Using 2D models is often impractical for large-scale EWS as they are computationally expensive when applied over hundreds or thousands of kilometres (Stokes et al., 2019; Stockdon et al., 2023). However, Quataert et al. (2025) tested the possibility of using XBeach 2D to define both erosion and flooding within a large-scale

**Table 1.** Large-scale (regional/national) level EWS, their hazards, hazard indicators, and running modes. The physics-based model, parametric formulae, and surrogate model only refer to the final processes used to define the hazards (e.g. runup, overtopping discharge, retreat) rather than the entire running method or model train

| Authors | Location | Hazard | Hazard Indicator | Running mode | Physics-hydro-morphodynamic model | Parametric formulae | Surrogate mode |
|---------|----------|--------|------------------|--------------|-----------------------------------|---------------------|----------------|
| Lane et al. (2008) | North England (UK) | Flooding | Overtopping discharge | Parametric formulae + Surrogate | – | Hedges and Reis (1998) | Pre-run matrices |
| Doong et al. (2012) | Taiwan | Flooding | Max. total water level | Physics-based model | SWAN + POM | – | – |
| Zou et al. (2013) | Southern England (UK) | Flooding | Overtopping discharge | Parametric formulae | – | Peng and Zou (2011) | – |
| Barnard et al. (2014) | California (USA) | Erosion | Shoreline retreat | Physics-based model | XBeach 1D | – | – |
| | | Flooding | Maximum total water depth + Flood extension | Physics-based model | XBeach 1D | – | – |
| Gaztelumendi et al. (2016) | Basque Coast (Spain) | Flooding | Maximum total water depth | Physics-based model | EUSKALMET | – | – |
| Lerma et al. (2018) | Aquitaine (France) | Erosion | Collision Regime + Eroded Volume | Parametric formulae | – | Stockdon et al. (2006) Splinter et al. (2014) | – |
| Stokes et al. (2019, 2021) | Southwest England (UK) | Flooding | Overtopping discharge + Wave height and freeboard | Parametric formulae | – | De Waal and van der Meer (1992); Stockdon et al. (2006); Poate et al. (2016); EurOtop (2018) | – |
| Stockdon et al. (2023) | Atlantic Coast + Gulf of Mexico (USA) | Erosion | Total water level + Sallenger's regimes | Parametric formulae | – | Stockdon et al. (2006) | – |
| | | Flooding | | Parametric formulae | – | Stockdon et al. (2006) | – |
| Turner et al. (2024) | Southwest and Southeast Coasts (Australia) | Erosion and Flooding | Erosion hazard scale + Flooding hazard scale | Parametric formulae | – | Stockdon et al. (2006); Harley et al. (2009); Armaroli et al (2012); Leaman et al. (2021); Leaman (2022) | – |
| Gorski et al. (2025) | Atlantic Coast + Gulf of Mexico (USA) | Erosion (dune impact) | Dune impact | Physics-based model | XBeach 1D | – | – |
| Quataert et al. (2025) | East Coast + Gulf of Mexico (USA) | Erosion and Flooding | Sallenger's regimes | Physics-based model | XBeach 2D | – | – |

EWS. In this test case, the computational expense was minimised by optimising the domain size, grid resolution and model settings.

Large-scale EWS can be devoted to specific coastal types, such as areas protected by coastal defences (e.g. Lane et al., 2008; Zou et al., 2013), or to a large variety of morphologies (e.g. Barnard et al., 2014; Stockdon et al., 2023; Turner et al., 2024), which can include embayed beaches, sandy coasts, gravel beaches and engineered areas (e.g. Stokes et al., 2019, 2021). None of the recently developed regional or national-scale EWS uses a surrogate mode for the final hazard definition. Only the older approach by Lane et al. (2008) makes use of pre-run matrices to avoid large computational expense, which was incompatible with the timely issuance of warnings. To avoid the use of surrogate modes and to ensure timely warnings, the regional/national-scale EWS have simpler approaches than hotspot-level systems, such as parametric equations and repeated use of XBeach 1D alongshore. Systems that do not require a surrogate and can operate in real time have the advantage of being able to incorporate the latest available coastal morphology at no extra computational cost, thereby improving the accuracy of the prediction (Stokes et al., 2021). To minimise computational effort and categorise the predicted extent, location and severity of erosion and flooding, large-scale EWS can use tools such as threshold-based decision tree models (e.g. Turner et al., 2024), which translate the modelled nearshore and inshore water levels and waves into hazards.

### EWS at the hotspot/local level

EWS developed for hotspots or the local level O(100–1,000 m) mostly have the same background architecture as regional/national scale EWS when using physics-based models. This means they rely on a series of global/regional meteorological models, nearshore hydrodynamic models, and on hydro-morphodynamic local and detailed models to define the hazard (mostly XBeach), either in 1D (transects) or 2D (coastal areas) applications (Table 2). These systems are rarely devoted solely to erosion hazards (e.g. Harley et al., 2011), typically integrating joint forecasts for storm-induced erosion and flooding using various indexes. Initially developed systems mostly used XBeach 1D (e.g. Vousdoukas et al., 2012; Souza et al., 2013; Valchev et al., 2014; Harley et al., 2016). In order to achieve greater detail at the hotspot level and fuller understanding of potential hazards, more complex and computationally expensive models began to be used. By the mid-to-end of the 2010s, there was an evolution towards using XBeach 2D coupled with a Bayesian Network to surrogate the model results (e.g. Poelhekke et al., 2016; Barquet et al., 2018; Plomaritis et al., 2018; Valchev et al., 2018; Garzon et al., 2023b, 2024). Bayesian Networks fed and trained by hundreds or thousands of previous model runs, and acting as a surrogate for the models, eliminate the need to run detailed and computationally expensive morphodynamics models within the EWS (van Dongeren et al., 2018). In a few cases, the 2D approaches do not use a surrogate mode (e.g. Seok and Suh, 2018; Turner et al., 2024), operating fully in real time.

Local/hotspot EWS are primarily used for flooding (overwash/overtopping), with the hazard being defined by parametric formulae (Poseiro et al., 2014; Sabino et al., 2018; Fortes et al., 2020; Chondros et al., 2021; Chatzipavlis et al. submitted) or physics-based models such as XBeach 1D (Harley and Ciavola, 2013; Sànchéz-Artús et al., 2025), TELEMAC (Bolle et al., 2018; Jäger et al., 2018), SWASH (Idier et al., 2020, 2021; Zózimo et al., 2025) or CSHORE (Papadimitriou et al., 2024). Several EWS devoted to flooding use surrogates that minimise computational effort while guaranteeing timely results. These include Bayesian Networks (e.g. Bolle et al., 2018; Jäger et al., 2018), Gaussian Processes models (e.g. Idier et al., 2020, 2021), Integrated Power Law Approximation (e.g. Merrifield et al., 2021) and Artificial Neural Networks (e.g. Zózimo et al., 2025) (Table 2).

The hotspot-level EWS have mostly been developed for sandy coasts, but several also include areas with coastal defences and ports (e.g. Bolle et al., 2018; Valchev et al., 2018; Fortes et al., 2020; Idier et al., 2021; Merrifield et al., 2021; Chondros et al., 2021; Zózimo et al., 2025), urban beaches (Biolchi et al., 2022; Garzon et al., 2023b, 2024; Sànchés-Artús et al., 2025; Chatzipavlis et al., submitted) and even gravel beaches and salt marshes (Jäger et al., 2018). This demonstrates the high diversity of applicability of the existing systems.

The definition of hazards in the most recently developed EWS at the hotspot-scale includes systems with more complex physics-based models, either alone or coupled with surrogates (e.g. XBeach 2D + Bayesian Network), simpler modelling approaches (e.g. XBeach 1D, CSHORE), or parametric formulae (e.g. for runup and overtopping). The choice of each EWS type and architecture depends on the complexity of the study area and the existing hazards, as well as on the data availability and accuracy for each hotspot.

## Storm impact indicators and warning levels

Indicators provide a simple way of presenting complex coastal hazard data and information (Carapuço et al., 2016; Ferreira et al., 2017). These indicators are commonly used to express potential storm impacts in coastal areas (Nguyen et al., 2016), and thresholds can be used to determine the potential damage associated with morphological changes or water levels (Ciavola et al., 2015). They are thus an important tool for providing clear information to first responders, stakeholders and the general public. The outputs of physics-based models and parametric formulae can be used to define the level of flooding or erosion hazard at a given coastal site, or even at a regional or national scale. These outputs are often used alongside pre-defined thresholds to better identify and categorise hazard levels. However, a wide variety of indicators are used to describe hazards associated to coastal storms (Ferreira et al., 2017) and this variety was also evident on the EWS analysed (Tables 1 and 2). The thresholds used and the categorisation of hazard levels vary greatly from EWS to EWS and are often site-specific. There is no universal set of indicators and levels that are consistently adopted and utilised globally in EWS for coastal hazards on wave-dominated coasts. By contrast, several EWS for inundation (not considering wave action) have similar indicators, such as flood depth or total water level, with broad applicability at a national or global scale (see Sweet et al., 2018, for example).

### Flooding

Nine different indicators (or groups of indicators) were used in works that developed EWS for flooding in coastal areas. This high number of indicators reflects the diversity of choices among authors and sites. Nevertheless, two indicators stand out, as they were used in different works and coastal areas:

- Total water level, used for both the hotspot scale (e.g. Vousdoukas et al., 2012; Fortes et al., 2020; Merrifield et al., 2021; Chatzipavlis et al., submitted) and for the regional/national scale (e.g. Doong et al., 2012; Barnard et al., 2014; Gaztelumendi et al., 2016; Stockdon et al., 2023; Turner et al., 2024). Chatzistratis et al. (2025)

**Table 2.** Local-scale EWS and EWS prototypes, their hazards, hazard indicators, and running modes. The physics-based model, parametric formulae and surrogate model only refer to the final processes used to define the hazards (e.g. runup, overtopping discharge, retreat), rather than the entire running method or model train. Some of the developed systems are grouped by site and similarity, and often represent the evolution of the same EWS

| Authors | Location | Hazard | Hazard indicator | Running mode | Physics-hydro-morphodynamic model | Parametric formulae | Surrogate mode | Coastal morphology |
|---|---|---|---|---|---|---|---|---|
| Harley et al. (2011); Harley and Ciavola (2013); Harley et al. (2016); Biolchi et al. (2022) | Several sites at the Emilia-Romagna Coast (Italy) | Erosion | Safe corridor width + Building-waterline distance | Physics-based model | XBeach 1D | -- | – | Sandy + Urban |
| | | Flooding | | Physics-based model | XBeach 1D | – | – | Sandy + Urban |
| Vousdoukas et al. (2012) | Faro Beach (Portugal) | Erosion | Dune foot retreat | Physics-based model | XBeach 1D | – | – | Sandy + Urban |
| | | Flooding | Total water level | Physics-based model | XBeach 1D | – | – | Sandy + Urban |
| Souza et al. (2013) | Sefton Coast (UK) | Erosion | Total water level + Wave height | Physics-based model | XBeach 1D | – | – | Sandy |
| | | Flooding | | Physics-based model | XBeach 1D | – | – | Sandy |
| Poseiro et al. (2014); Sabino et al. (2018); Fortes et al. (2020) | Azores harbours (Portugal) | Flooding | Overtopping discharge + Total water level | Parametric formulae | – | EurOtop (2007), Coeveld et al. (2005) and several runup formulae | – | Defenced |
| Valchev et al. (2014) | Kamchia-Shkorpilovtsi (Bulgaria) | Erosion | Beach face retreat | Physics-based model | XBeach 1D | – | – | Sandy |
| | | Flooding | Flood depth-velocity product + Max. wave runup extension | Physics-based model | XBeach 1D | – | – | Sandy |
| Poelhekke et al. (2016); Plomaritis et al. (2018); Garzon et al. (2023b, 2024)* | Faro Beach (Portugal) | Erosion | Vertical erosion + Horizontal distance + Erosion impact index | Physics-based model + Surrogate | XBeach 2D | – | Bayesian Network | Urban |
| | | Flooding | Overtopping discharge | Physics-based model + Surrogate | XBeach 2D | – | Bayesian Network | Urban |
| Barquet et al. (2018) | Äspet (Sweden) | Erosion | Not stated | Physics-based model + Surrogate | XBeach 2D | – | Bayesian Network | Sandy |
| | | Flooding | | Physics-based model + Surrogate | XBeach 2D | – | Bayesian Network | Sandy |
| Bolle et al. (2018) | Zeebruge harbour (Belgium) | Flooding | Flood depth | Physics-based model + Surrogate | TELEMAC | – | Bayesian Network | Defenced |
| Jäger et al. (2018) | Wells-next-the-Sea (UK) | Flooding | Flood depth + Max. flood depth-velocity product + Overtopping discharge | Parametric formulae + Physics-based model + Surrogate | TELEMAC | EurOtop (2007) | Bayesian Network | Gravel and Sand + Saltmarsh |

(*Continued*)

**Table 2.** (*Continued*)

| Authors | Location | Hazard | Hazard indicator | Running mode | Physics-hydro-morphodynamic model | Parametric formulae | Surrogate mode | Coastal morphology |
|---|---|---|---|---|---|---|---|---|
| Seok and Suh (2018) | Haeundae Beach (South Korea) | Erosion | Wave height | Physics-based model | XBeach 2D | – | – | Sandy |
| | | Flooding | | Physics-based model | XBeach 2D | – | – | Sandy |
| Valchev et al. (2018) | Varna Bay (Bulgaria) | Erosion | Morphological changes | Physics-based model + Surrogate | XBeach 2D | – | Bayesian Network | Sandy + Defenced |
| | | Flooding | Wave runup | Physics-based model + Surrogate | XBeach 2D | – | Bayesian Network | Sandy + Defenced |
| Idier et al. (2020, 2021) | Gâvres (France) | Flooding | Max. water volume + Water discharges + Flood depth | Physics-based model + Surrogate | SWASH | – | Gaussian Processes metamodel | Sandy + Defenced |
| Chondros et al. (2021) | Rethymno (Greece) | Flooding | Flood depth | Parametric formulae | – | EurOtop (2018) | – | Defenced |
| Merrifield et al. (2021) | Imperial Beach (USA) | Flooding | Total water level | Physics-based model + Surrogate | SWASH | – | Integrated Power Law Approximation | Sandy + Defenced |
| Papadimitriou et al. (2024) | Pyrgos (Greece) | Flooding | Overtopping discharge | Physics-based model | CSHORE | – | – | Sandy |
| Turner et al. (2024) | Several sites at the Australian Coast | Erosion | Dune impact exposure + Cumulative wave energy + Dune stability factor + Safe corridor width | Physics-based model | XBeach 2D | – | – | Sandy |
| | | Flooding | | Physics-based model | XBeach 2D | – | – | Sandy |
| Zózimo et al (2025)* | Figueira da Foz harbour (Portugal) | Flooding | Overtopping discharges | Physics-based model + Surrogate | SWASH | – | Artificial Neural Networks | Sandy + Defenced |
| Chatzipavlis et al. (submitted) | Poeto Beach (Italy) | Flooding | Total water level | Parametric formulae | – | Stockdon et al. (2006) | – | Urban |
| Sánchez-Artús et al. (2025) | Barcelona (Spain) | Flooding | Percentage of flooded area | Physics-based model | XBeach 2D | – | – | Urban |

*The developed systems were integrated on the HIDRALERTA EWS (see Fortes et al. 2020 for a description of HIDRALERTA).

state that the majority of the coastal flood EWS in Europe use total water level as an indicator without considering flood characteristics (e.g. flood extent, depth, velocity) and impacts. According to the authors, this is inconsistent with user needs and the existing policy and legal framework, which limits the wider application of the existing EWS (for example, at a pan-European level).

- Overwash/overtopping discharge, used at both the hotspot (e.g. Plomaritis et al., 2018; Fortes et al., 2020; Garzon et al., 2023a, b, 2024; Papadimitriou et al., 2024; Zózimo et al., 2025) and regional/national (e.g. Lane et al., 2008; Zou et al., 2013; Stokes et al., 2021) scales.

Flood depth is also commonly used (e.g. Bolle et al., 2018; Jäger et al., 2018; Idier et al., 2021; Chondros et al., 2021), either on its own or alongside other parameters, such as flow velocity. Some authors have used other indicators (e.g. building-waterline distance, maximum runup position, safe corridor width, percentage of flooded area, wave height and freeboard) for similar EWS.

There is no universally accepted number of flood hazard levels. Approaches range from a Yes/No assessment (e.g. Stockdon et al., 2023) to a maximum of five levels (e.g. Bolle et al., 2018; Idier et al., 2021). The majority of studies use three or four levels, distinguishing between them using a generic traffic light approach. There is also no commonly used set of thresholds for splitting hazard levels, with each author defining their own for the studied site or region.

For the simple Yes/No approach flooding potential, the most commonly used method to differentiate between hazard levels is Sallenger's storm regime (Sallenger, 2000), which categorises them as swash, collision, overwash or inundation according to the relationship between water levels and morphology. However, this approach does not differentiate between levels of overwash potential (the difference between the total water level and the highest morphology; Matias et al., 2012, 2016) or inundation potential. Consequently, a minimum potential for flooding is categorised in the same way as a major one. A ranking of hazard levels according to their flooding potential, which can be used by a large number of stakeholders worldwide, has yet to be achieved.

Several authors who use overtopping discharges as an indicator refer to the limits defined in EurOtop (2018), which are mostly applied to coastal defences or urbanised coasts. Garzon et al. (2023a,b) proposed a unified set of mean overtopping/overwash discharge limits, that can be used globally, split into four warning levels (Table 3), ranging from No Impact to High Impact, and for three types of exposed elements (pedestrians, urban components and buildings and vehicles). This is the most recent and comprehensive work on the unified use of overtopping discharge as an indicator of coastal flooding hazard.

**Table 3.** Proposed unified mean discharge limits (litres per second per metre) to be implemented in EWS to define the impact on three types of receptors (Garzon et al., 2023a)

| Coastal Receptor | No impact | Low impact | Moderate impact | High impact |
|---|---|---|---|---|
| Pedestrians | <0.03 | [0.03–0.1[ | [0.1–1.0[ | > = 1.0 |
| Urban components & buildings | <1.0 | [1.0–2.5[ | [2.5–20[ | > = 20.0 |
| Vehicles | <0.5 | [0.5–1.5[ | [1.5–5[ | > = 5.0 |

## Erosion

The hazard indicators for coastal erosion are highly diverse, with eight types identified in the analysed EWS. No single indicator is widely used by several authors/EWS. However, several studies use dune foot retreat or beach face retreat (e.g. Vousdoukas et al., 2012; Barnard et al., 2014; Valchev et al., 2014; Harley et al., 2016; Plomaritis et al., 2018; Garzon et al., 2024; Turner et al., 2024). These parameters are often combined with vertical erosion and/or distance to a given element (e.g. a building or dune) to create an indicator mostly used at hotspot scale (e.g. Harley et al., 2016; Plomaritis et al., 2018; Garzon et al., 2024; Turner et al., 2024). Other indicators used for coastal erosion include the total water level (associated with Sallenger's regimes), wave height, eroded volume and cumulative wave energy. The number of hazard levels considered by different EWS ranges from Yes/No to a maximum of four levels. Authors commonly use a traffic light (or similar) approach to distinguish between hazard levels. The only threshold used in multiple studies is the collision regime limit, which indicates potential dune erosion. All other proposed thresholds between hazard levels are site- or region-specific, and no proposal has been made for a unified use of limits.

One study (Turner et al., 2024) uses a method developed by Leaman et al. (2021), for regional and national scales, which combines an erosion indicator based on beach face and dune retreat with a flooding hazard indicator based on total water level. This yields 16 combined flood/erosion hazard levels, categorised by morphological changes and storm regimes (using the Sallenger's storm impact scale). This method, the Storm Hazard Matrix, can be used globally to screen vast areas for each storm condition.

## Accuracy and usability of EWS

The vast majority of developed EWS have a very limited accuracy assessment (see Table 4), primarily due to the difficulty of finding large data sets with reliable flood or erosion indicators measurements (e.g. flood depth or extent, overtopping discharge, dune retreat or beach vertical erosion). Most studies that performed a quantitative assessment against field data used a small number of storms and a limited number of sites/transects. An exception is the work of Turner et al. (2024), which used 43 storms across 36 sites to evaluate hazard levels and the impact on dunes. Several studies used visual observations or qualitative indicators of hazard or impact to compare with predicted hazard levels. Others only evaluated the degree to which the surrogate mode was accurate compared to the original model (see Table 4). Existing assessments indicate a level of agreement ranging from "reliable" to "good" when the assessment is qualitative, and a generic accuracy range of 50–85% when predicting a given hazard level (Table 4). Some studies state that most hazard level deviations are within one hazard level (see, e.g., Garzon et al., 2024). When evaluating the simple occurrence (or not) of a given hazard (a Yes/No approach), the accuracy rises to values of around 80–95%, which clearly shows that these systems can at least indicate whether a hazard is likely or not. However, there is a clear need for intensive and continuous field observations during storm events to enable direct comparison of predicted and measured indicators. Without this, it will not be possible to determine if the developed EWS have a high level of reliability.

Based on the existing accuracy assessments, which are still incomplete and limited in scope, it can be concluded that the EWS developed so far can predict and anticipate the occurrence of a flood or erosion hazard (Yes/No approach) with a high

**Table 4.** Accuracy assessment methods and results for coastal erosion and flood hazards EWS. The percentages represent the level of accuracy obtained. BSS – Brier Skill Score

| Authors | Method | Accuracy | Hazard |
|---|---|---|---|
| Harley et al. (2011) | Field data (limited) | Reasonable dune retreat agreement | Erosion |
| Souza et al. (2013) | Field data (limited) | BSS Good agreement | Erosion |
| Barnard et al. (2014) | Field data (limited) | Positive reproduction of profile evolution | Erosion |
| | | Enough to support EWS | Flood |
| Poelhekke et al. (2016) | Modelled results* | 81%–88% | Erosion/Flood |
| Valchev et al. (2018) | Field data (limited) | Reliable (variable BSS) | Erosion/Flood |
| Chondros et al. (2021) | Modelled results | Minimal differences in water levels | Flood |
| Idier et al. (2021) | Observations (Yes/No) | 92% | Flood |
| Stokes et al. (2021) | Visual observations (Yes/No) | 97% (between low/high hazard) | Flood |
| | Field data (limited) | 80% (Yes/No) 50% (Correct hazard level) | Flood |
| Zózimo et al. (2021) | Damage observations (limited) | Agreement | Flood |
| Garzon et al. (2023b) | Visual and damage observations | 76% Pedestrians 87% Vehicles 84% Buildings | Flood |
| Garzon et al. (2024) | Modelled results* | 64%–72% | Erosion |
| Turner et al. (2024) | Field data (43 storms and 36 sites) | 76% Hazard level 86% Dune Impact | Erosion/Flood |
| Chatzipavlis et al. (submitted) | Field data (limited) | Strong agreement | Flood |
| Sánchez-Artús et al. (2025) | Field data (limited) | 82% | Flood |

*Assessment of the Bayesian Network accuracy as a surrogate of the model.

probability of correctness. When defining the expected hazard level, which is often categorised into three or four levels, the EWS correctly identify the hazard level in at least half of the cases. Most errors differ just one level from the level effectively observed. This suggests that the existing EWS can distinguish between high and low hazard levels, but may struggle to accurately define intermediate hazard levels.

## Limitations and future developments

### Local versus large scale approaches

EWS developed for wave-dominated open coastal areas (erosion and flooding) were mostly designed and applied at a local scale (site-specific/hotspot), with fewer examples of use at a larger regional or national scale. While local-scale, high-resolution EWS are fundamental to correctly predicting hazards and risks, issuing meaningful warnings to end users and the population, there is still a need to develop large-scale systems that provide an initial screening of such hazards and risks for larger areas. These systems are useful for identifying all hotspots (for a given storm or set of storms), for which detailed approaches will then be required. They can also be a relevant solution for regions and countries with restricted data availability that cannot implement data-rich based EWS. Global wave databases (see Fanti et al., 2023, for a review and error analysis), global coastal topobathymetric databases (see Pronk et al., 2024 and Fanti et al., 2025, for a review and error analysis) and global socioeconomic indicators (see Athanasiou et al., 2024) can be used as a first source of information for such large scale EWS.

### Data assessment

A major limitation to the more extensive and accurate use of EWS is the lack of high-quality hazard and impact measurements for low-frequency, high-impact events that can be used to validate existing and new EWS. Even in so-called "data-rich environments," the level of testing and validation is relatively modest (see Table 4). For most coastal areas of the world, there is no in situ data on the impact of storms, which hinders the full development of EWS.

As up-to-date topobathymetric data are not often available (Matheen et al., 2021), most existing EWS for coastal erosion and flooding use synthetic or representative morphological data from past surveys. However, these morphologies may not accurately represent the actual pre-storm beach morphology, thereby increasing the uncertainty of the final result. Therefore, EWS should include accurate and updated morphological data, and ensure a regular data assimilation into the modelling process. Regularly updating the topobathymetric conditions in this way is critical to improving the performance of EWS.

To minimise the limitations of the availability and updating of morphological data, particularly at the local scale, it is important to establish monitoring programmes that can rapidly transfer and assimilate data into the modelling system (Leaman et al., 2021; Stokes et al., 2021). Such programmes may include automated beach profiling, beach cameras, satellite-derived topobathymetry and data from unmanned aerial vehicles (UAV) or LiDAR surveys (airborne or stationary). In remote areas or where field data is difficult to obtain, global models can be a solution for mitigating data scarcity and validating EWS. Emerging coastal altimetry datasets have the potential to enhance model performance at the coast

(Apecechea et al., 2023). Furthermore, global coastal flooding maps produced using available open data sources (including global models) are approaching the resolution of local flooding maps (Baart et al., 2024). However, errors in the existing global datasets for coastal topography and bathymetry (Pronk et al., 2024, Benito et al., 2025, Fanti et al., 2025) and storm waves (Fanti et al., 2023; Benito et al., 2025) must be taken into account, as they are likely still large enough to prevent accurate results at the local (hotspot) scale.

### Morphodynamic feedback

Morphodynamic feedback is highly relevant in determining the final impact of a storm. For example, the lowering or breaching of a dune can result in further overwash, leading to higher flood velocities, extensions or depths. Several methods used by EWS to simulate wave-induced flooding or dune erosion neglect these morphodynamic feedbacks. However, if these feedbacks are strong enough, the current methods may be inaccurate and alternatives may be required (Garzon et al., 2023a,b). Consequently, EWS should consider morphodynamic feedback as storms occur, as well as their alongshore and cross-shore heterogeneities (Barnard et al., 2014; Papadimitriou et al., 2024), which are most commonly not included. This would require the use of full 2D models to simulate both cross-shore and alongshore processes, incorporating the morphological evolution into the modelling process.

### Computational time and surrogate modes

Process-based models (such as XBeach 2D) are still computationally expensive to run at regional or national scales (Stokes et al., 2019, 2021), which often prevents their use in forecasting and issuing warning (Idier et al., 2021). Therefore, empirical formulations that compute wave runup, overtopping discharge, or shoreline/dune retreat have been used to forecast coastal flooding and erosion at large-scale applications. Another solution to minimise the problems associated with high computational time, particularly for hydrodynamic and morphodynamic models, is to use surrogate modes to predict hazards. Apart from being fast, an additional benefit of surrogate modes (e.g. Bayesian Networks) is that they can frequently incorporate probability distributions of the hazard, whereas numerical models typically only provide deterministic outcomes (Garzon et al., 2023b). There is a wide variety of surrogate types, some of which are used in existing applications, such as Bayesian Networks (e.g. Jäger et al., 2018), metamodels (e.g. Idier et al., 2021) and Artificial Neural Networks (e.g. Zózimo et al., 2025). An adequate surrogate should include the maximum possible number of model runs (hundreds to thousands) to be stored in the surrogate and/or used to train it. The surrogate results should always be validated against an independent number of model results and, if possible, against a set of field observations. As comprehensive datasets representative of the extreme and episodic nature of storm-induced hazards are unavailable for most coastal areas, most of the developed Bayesian Networks (and probably other surrogates) are trained using data generated by process-based models (Callens et al., 2022). As high-resolution data from remote sensing and monitoring programmes becomes more widely available, there will be more opportunities to combine data-driven and physics-based models, improving the accuracy of predictions. Hybrid models, which combine data driven and physics-based models, are likely to become more prevalent in the future (Haddad, 2025). As an alternative approach to minimising the computational time required by complex process-based models, models with lower computational demand can be used. Examples include SFINCS (Super-Fast Inundation of Coasts; Leijnse et al., 2021) for flooding, and alongshore repeated XBeach 1D for erosion. These models are fully compatible with the timely warning needs of the EWS.

In the future, improving EWS and reducing their computational expense could be achieved by replacing parts of the modelling process with artificial intelligence/machine learning techniques such as neural networks. However, these techniques must be properly trained to make accurate hazard predictions (Simmons and Splinter, 2022; Ibaceta and Harley, 2024; Senechal and Coco, 2024; Turner et al., 2024). The validity of these approaches depends on the underlying data, as a large quantity of field observations is necessary to properly train them and to achieve good accuracy. In coastal domains, large, continuous datasets on coastal flooding or erosion are not yet common, which limits the current use of data-driven approaches.

### From hazards to risk (and people)

Most of the EWS described stop at hazard determination, ranking or mapping, and only a few include the risk to assets, goods or people. Furthermore, at least at the European level, there is a low coherence between the needs of potential end users and the characteristics of the available EWS, with the latter lacking useful information such as details of associated damage (Chatzistratis et al., 2025). EWS will only be effective if they include an assessment of the risks and actions tending to minimise the impact of storms in coastal areas (Ciavola et al., 2015). The key to successfully developing a comprehensive EWS lies in cooperating with the communities of users (WMO, 2022b). However, there is still a tendency to focus either on the technical aspects of natural hazards (the aim of the current review) or on social aspects alone, with little integration between the two (Barquet et al., 2018). To achieve this integration, a higher level of engagement from end users and stakeholders is needed, including their participation in the co-design of the EWS and co-creation of alerts and responses to warnings. To ensure the reliability of EWS, it is also important to correctly align the warning thresholds with the end users' needs (Zózimo et al., 2021; Chatzistratis et al., 2025). To this end, the specific requirements of each user must be identified in the user requirements plan, and customised forecast and warning interpretive products should be developed for each local area or region, in consultation with the local users (WMO, 2022b).

In future, an important aspect to consider is how human behaviour varies over time or in response to a given hazard and how this affects overall risk computation. The behavioural components of risk assessment can be incorporated into the EWS by adjusting the level of human occupation, according to the hour, day or season, for example (Garzon et al., 2023b). Agent-based models can be used to assess the human response in real time. Such models have been employed to examine adaptation to climate change in coastal regions (see Noia, 2022, for a compilation of studies), including defining the vulnerability and resilience of coastal areas to climate change and extreme events (e.g. Tierolf et al., 2023; Roukonis et al., 2025). The use of such models in coastal zones will probably be explored further in the near future (Noia, 2022). One possible application is to incorporate them into EWS to understand the interaction between hazard predictions and human behaviour, thus enabling the future creation of risk maps adapted to human responses.

## Conclusions

Current EWS for coastal erosion and/or flooding hazards on wave-dominated open coasts are predominantly designed for local-scale applications, offering high-resolution predictions for specific sites. While these are important, there is still a pressing need to develop large-scale systems that can provide a broad screening of hazards across vast regions, particularly in areas where data is scarce. Such systems could act as a preliminary filter to identify critical hotspots requiring detailed local assessments. Furthermore, significant heterogeneity exists in the indicators and thresholds used by different EWS, reflecting site-specific approaches and the preferences of individual authors. There is still no universally accepted set of coastal flooding or erosion indicators and hazard levels. This lack of standardisation hinders the development of universally applicable systems and complicates cross-regional comparisons.

EWS demonstrate strong performance (80–95% accuracy) in binary hazard detection, i.e. the occurrence or non-occurrence of the hazard. This suggests their potential utility in alerting the public and end users to imminent storm impacts. However, while current EWS can effectively distinguish between low and high hazard levels, they often struggle to accurately identify intermediate hazard categories, which limits their precision. Most analysed EWS have undergone limited accuracy evaluation due to the scarcity of comprehensive field datasets containing measured hazard indicators. Many studies assessing the accuracy of EWS rely on qualitative comparisons or visual observations rather than rigorous field-based validation. Consequently, the lack of extensive and continuous field observations during storm events remains a significant obstacle to validating and improving EWS.

Most EWS rely on outdated or representative morphological inputs that may not accurately reflect pre-storm conditions, thereby increasing prediction uncertainty. Therefore, regular assimilation of updated topobathymetric data is essential to improve prediction reliability. Real-time monitoring programmes that use technologies such as beach cameras, UAV, LiDAR and satellite-derived bathymetry offer a promising way to address data limitations. Global datasets are also a potential source of information for this purpose. However, these datasets still contain significant errors that must be accounted for, particularly at hotspot scales.

The computational time required for physics-based models is still a major issue when it comes to issuing hazard warnings in a timely manner. This issue is often resolved by using modelling simplifications or surrogates. Artificial intelligence techniques offer a promising way to reduce computational demands and improve predictive accuracy. However, these techniques are only effective if large and representative field datasets are available, and these are currently lacking in most coastal regions.

Most of the analysed EWS focus solely on hazard detection, with limited integration of the risks to assets or populations. Effective systems must incorporate risk assessments and mitigation strategies, tailored to needs of end users. This requires active stakeholder engagement in the co-design and co-creation of warning systems and hazard thresholds.

**Open peer review.** To view the open peer review materials for this article, please visit http://doi.org/10.1017/cft.2026.10026.

**Data availability statement.** Data availability is not applicable to this article as no new data were created or analysed in this study.

**Acknowledgements.** Icons in Figure 1 were designed by Freepik. Images on Figure 2 and maps on Figure 3 were generated using Microsoft Copilot, an AI-powered assistant developed by Microsoft.

**Author contribution.** Óscar Ferreira is the sole contributor to this paper.

**Financial support.** This work was supported by the Portuguese Foundation for Science, under the projects LA/P/0069/2020 granted to the Associate Laboratory ARNET, and UIDP/00350/2020 granted to CIMA (https://doi.org/10.54499/UIDP/00350/2020).

**Competing interest.** None.

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
