## [Reviewer Report]

I found the manuscript “EARLY WARNING SYSTEMS FOR STORM-INDUCED COASTAL FLOODING AND EROSION” an informative synthesis on the various developments in coastal erosion and flooding early warning systems published in particular over the past 15 years (focussing predominantly in Europe, USA and Australia). The author provides a summary of these systems, highlights limitations and suggests future directions.

One of the challenges I had with the review in its current form is its scope. The review does not clearly clarify the bounds of the review, which seems mainly limited to published academic literature. Inevitably with forecasting systems a lot of the developments are only reported in grey literature, so my feeling is that this review probably captures only a smaller fraction of the developments efforts that are out there. As a small example, some systems that are not included in the review are: (1) the Aquitaine coast in France (https://www.observatoire-cote-aquitaine.fr/Le-reseau-tempetes); (2) efforts in Indonesia, Pacific Island countries and Bangladesh as part of the WMO Coastal Inundation Forecasting Demonstration Project (CIFDP); (3) INCOIS in India (https://incois.gov.in/site/services/hwa.jsp). I don’t see it necessarily as a problem that they are not included, but I think it should clearly state this limited scope at the outset.

Likewise, the term “coastal erosion and forecasting EWS” could be more clearly defined to help understand the scope. Is it limited to sandy coastlines? Likewise is it limited to extratropical coastlines (where waves dominate?) If not, then it appears to be missing key systems such as wave-driven coastal flooding on coral reef coasts (e.g. doi.org/10.3389/fmars.2020.00199) and tropical cyclone inundation forecasting systems (e.g. WMO CIF-EWS).

I think with a tighter clarification of the scope of the review this would assist greatly in managing expectations and remove any confusions. I have marked my recommendation as “Major revision”, but probably falls in a category between Minor and Major.

Please see line-by-line comments below

Manuscript title

The title is a bit vague - perhaps clarify it is a review article in the title? And following comments above, consider adding “sandy coasts”?

Figures

As a general comment, the figures are a little underwhelming and not particularly informative. Figure 2a suggests a length scale of 100km (so a forecast every 50km), which seems a lot?

Line 89: provide a reference for the “storms on coastal areas will increase”. Storm frequency to climate change is complex and might not necessarily increase

Line106: UNISDR are relatively old references. There has been quite a lot of developments in the Early Warning Systems by the United Nations in the past 5 years which is not mentioned in this manuscript, for instance the Early Warning Systems for all action plan (2023-2027). I feel this should be included

Line 134: how do these 4 modules relate to the four pillars of EWSs as described by the UNDRR?

Line 143: are “coastal hazards” here specifically referring to metocean hazards? What about winds, hail etc? These are often responsible for the biggest insurance losses

Line 151-153: “EWS are even less used for complex coastal areas or near complex coastal structures, such as ports” I would question the validity of this statement. I would imagine that, given the risk to operations, most major ports worldwide would have their own internal EWS for extreme events. The industry group PIANC have been addressing this (for example, https://pianc.org.au/wp-content/uploads/2020/10/DavidTaylor_PredictiveModellingofExtremeEvents_PIANC-New-Technologies-May-2019Comp1.pdf). Perhaps just not available in the academic literature?

Line 161-163: “EWS that neglect the wave component when assessing coastal hazards were not considered, as they underestimate these hazards” This seems like a key statement about the scope of the review and could be expanded further. I presume that this means that tropical cyclone inundation forecasting systems are subsequently neglected?

Line 186: The MICORE project needs to be explained here as it is the first time it is mentioned (it is explained in the next paragraph, which is a bit out of order)

Line 202-203: What does “fully operational” mean here? The WMO has some careful guidelines about operational coastal inundation forecasting systems (see WMO “Guidelines on Implementation of a Coastal Inundation Forecasting–Early Warning System”). I think this could be clarified

Line 230: typo here “jointlyr”

Section 5.2. It seems like a limitation here is also the availability of data to run an EWS even at the most basic level (e.g. topographic data). This could be considered a limitation too?

---

## [Reviewer Report]

Review of ‘Early Warning Systems for Storm-Induced Coastal Flooding and Erosion’ by Oscar Ferreira.

The review paper by Ferreira describes existing prototype and operational early warning systems for coastal erosion and flooding on coastline exposed to breaking waves. Overall the review is well-written and appears to be comprehensive. I have several comments which I would like to see addressed before publications.

Main concerns:

- Users and their needs are mentioned several times (Line 189, 358, 563) but there are limited details on what these are. Could you provide some examples (preferable from peer-reviewed or grey literature) of what makes an EWS user-friendly, or what users do/do not want from EWSs?

- The introduction needs to be clearer that this review only focuses on wave-dominated coastlines and does not consider EWS for flooding in sheltered areas like harbours and embayments, nor estuaries.

o Could consider changing the title to include “on open coasts”.

o At Line 157 could amend to “storm-induced coastal flooding and erosion on wave-dominated coastlines like sandy beaches”.

o At Line 162 “they underestimate these hazards” is only true if using to assess flooding on a wave dominated coastline. EWS without waves can accurately predict flooding in estuaries and other locations sheltered from swell waves (e.g. https://doi.org/10.3389/fmars.2022.1073792).

o At Line 164, I don’t get the why “focus primarily on hazard assessment and less on warning, communication and preparedness actions” justifies not including EWS for estuaries and compound flooding. It is fine to exclude these, I just don’t understand how these aren’t more or less about “hazard assessment” and “warning, communication, and preparedness actions”. Suggest revising this sentence.

- I think the framing around SLR making EWSs more relevant needs to be re-considered (e.g., Line 77, 92, 106). Firstly, many of the warning systems are based on parameters that are insensitive to SLR (e.g., wave height, cumulative wave energy, wave setup - see Line 409). Secondly, whilst total water level is expected to increase with SLR, my reading of the literature is that many forecast systems have not explicitly incorporated this – e.g., Stokes et al. 2021; Turner et al. 2024. If SLR is a key consideration it would be worth adding discussion about the need to refine parameters or thresholds as sea levels rise (and wave climate changes), otherwise make the scope about forecasting in the present (assumed stationary) climate.

Other comments:

- Line 97: Need reference for changes in climate variability.

- Line 106: Perhaps worth noting that EWS are not necessarily effective for all adaptation actions – for example if community retreats from coast may not need an EWS anymore.

- Line 230: Typo: ‘jointly’

- Section 3.1: It would be worth including discussion that impact-based flood thresholds exist for non-wave-domintaed coasts and are used operationally. If we had these for erosion we may have more consistency around the indicators used. The NOAA NOS and NWS thresholds are now ubiquitous in studies of locations where these are defined. See: https://www.tidesandcurrents.noaa.gov/publications/techrpt86_PaP_of_HTFlooding.pdf and citations thereof.

- 369: This sentence is a bit confusing. Can you just say “some authors”?

- Figure 1: I am unsure what the journal’s policy regarding the use of AI is, but using generative AI seems unnecessary here when many similar and suitable real photos exist under CC-BY (e.g., see Wikimedia Commons or previously published Open Access papers).

- Table 3: The closing square bracket is around the wrong way.

---

## [Editor Report]

I have read the comments of both reviewers and agree with their comments. Please address all of these coments with a particular emphasis on comments by Reviewer 1 on tightening clarification of the scope of the review and including a wider breadth of forecasting systems not included in the review. Both reviewers in fact comment the need to more clearly define the scope of the review and I agree that this will improve the paper. Clarifying whether the review is limited to sandy coastlines, if it considers extra-tropical caosts, and whether sheltered areas like harboursembayments/estuaries are meant to be omitted would be very worthwhile..

---

## [Reviewer Report]

I commend the author for taking the comments raised in the review into account. I am satisfied that the comments have been addressed.

The only remaining issue I feel is the title - I do not think “Open coasts” is quite what this review is, as it focuses on wave-dominated open coasts (a subset of open coasts). I therefore suggest a title change to: A REVIEW OF EARLY WARNING SYSTEMS FOR STORM-INDUCED COASTAL FLOODING AND EROSION ON WAVE-DOMINATED OPEN COASTS”

There are also a few spelling mistakes that I noticed -“which must be acknowledged” (Line 203) and “overwarsh” (Line 416). There may be others so I suggest a close check prior to publishing

---

## [Editor Report]

Thank you for addressing comments from both reviewers. 

Please address very minor comments from reviewer 2 regarding a title change and addressing spelling errors. Once that is done, the paper is ready to accept.